# Innovations, contestations and fragilities of the health system response to COVID-19 in the Gauteng Province of South Africa

**Laetitia C. Rispel** [1]*, **Carol Marshall**[2], **Busisiwe Matiwane** [1], **Immaculate Sabelile Tenza** [1]

**1** Centre for Health Policy & South African Research Chairs Initiative (SARChI), School of Public Health, Faculty of Health Sciences, University of The Witwatersrand, Johannesburg, South Africa, **2** School of Public Health, Faculty of Health Sciences, University of The Witwatersrand, Johannesburg, South Africa

* laetitia.rispel@wits.ac.za

**Data Availability Statement:** The documents analysed are available publicly. The interview data cannot be shared publicly because of the relatively small number of key informants who could be

## Abstract

### Background

Gauteng province, with 26.3% of South Africa's population, is the commercial and industrial powerhouse of the country. During the first epidemic wave in 2020, Gauteng accounted for 32.0% of South Africa's reported COVID-19 cases.

### Aim

The aim of this study was to describe the health system response to the COVID-19 pandemic during the first epidemic wave in Gauteng province and to explore the perspectives of key informants on the provincial response.

### Material and methods

Using an adapted Pandemic Emergency Response Conceptual Framework, this was a qualitative case study design consisting of 36 key informant interviews and a document analysis. We used thematic analysis to identify themes and sub-themes from the qualitative data.

### Results

Our case study found that Gauteng developed an innovative, multi-sectoral and comprehensive provincial COVID-19 response that aimed to address the dual challenge of saving lives and the economy. However, the interviews revealed multiple perspectives, experiences, contestations and contradictions in the pandemic response. The COVID-19 pandemic exposed and amplified the fragilities of existing systems, reflected in the corruption on personal protective equipment, poor data quality and inappropriate decisions on self-standing field hospitals. Rooted in a chronic under-investment and insufficient focus on the health workforce, the response failed to take into account or deal with their fears, and to incorporate strategies for psychosocial support, and safe working environments. The single-minded focus on COVID-19 exacerbated these fragilities, resulting in a de facto health system

identified. The Human Research Ethics Committee (Medical) of the University of the Witwatersrand (contact via https://www.wits.ac.za/ethics/human-research-ethics-committee-medical) has imposed restrictions because of the confidential and sensitive nature of the data. Contact: Iain.Burns@wits.ac.za.

**Funding:** The South African Research Chairs Initiative (SARChI) of the National Research Foundation in South Africa partially supported this study (Grant # 102219). The funding source was not involved in the design of the study, the data collection or analysis, in the interpretation of the results, or in the writing of the manuscript.

**Competing interests:** The authors declare no conflict of interest, financial or otherwise.

lockdown and reported collateral damage. The key informants identified missed opportunities to invest in primary health care, partner with communities and to include the private health sector in the pandemic response.

## Conclusion

Gauteng province should build on the innovations of the multi-sectoral response to the COVID-19 pandemic, while addressing the contested areas and health system fragilities.

## Introduction

Globally, the devastation of the Coronavirus disease 2019 (COVID-19) pandemic is reflected in close to 5 million deaths by 26 October 2021 [1]. The shocks to social and economic systems have exacerbated pre-existing inequities, fragilities and unsustainable practices [2]. Several factors have influenced country-specific innovations and responses to the COVID-19 pandemic. These include political leadership, legislative controls, previous experience of respiratory diseases, existing disaster or pandemic management plans, national health systems, and technology [3–6]. Notwithstanding exemplary practices in many countries, in January 2021, the Independent Panel for Pandemic Preparedness and Response highlighted "the [global] failures in the chain of preparedness and response" [3]: page 5.

The relentless COVID-19 pandemic resulting from the novel SARS-CoV-2 virus has underscored the importance of resilient and well-performing health systems, able to prevent new infections, respond to increased demand, provide safe and effective treatment and care, and prevent deaths [6–9]. The literature on COVID-19 reveals complex, yet innovative and at times contradictory country-level responses during the first wave of COVID-19 infections [5, 10–24]. The variations in country-level responses resulted in differential impact of the pandemic during the first epidemic wave, further shaped by diverse political, social, cultural, economic and health system contexts, the timing and range of government policies, quality of information systems, and the relative trade-offs between population health and the economic impact of interventions [6]. Some countries followed a holistic health system response that included increased budgets, investment in human resources and infrastructure, prioritisation of vulnerable individuals, access to mental health services, and research and development [6, 14, 20, 24].

Evidence suggests that China's centralised response of a stringent, national lockdown, additional health care resources, technology utilisation for contact tracing and to monitor population travel patterns, and the use of social media to disseminate government information, and to conduct public campaigns during the first wave limited the pandemic largely to Wuhan province [24]. Austria, Germany, and Switzerland's early response of clear testing strategies, contact tracing, case containment and sufficient state capacity for rapid policy implementation were effective in preventing the spread of the virus, protecting the lives of their citizens and reducing economic damage [14]. Finland's response combined Emergency Powers legislation, increased critical care bed capacity, medical personnel increases, and private sector involvement [23]. Australia's first wave experience suggests that the federal government was able to ensure strong central coordination through consultation with and support of the state governments to implement the COVID-19 pandemic response, thereby reducing its impact [17].

In contrast, despite increases in health care resources in Italy [12], Spain [16], the United Kingdom [15], and the United States [13], a combination of suboptimal leadership, poor or conflictual intergovernmental relations or collaboration, and fragmentation hindered the

implementation of a comprehensive pandemic response. This resulted in geographical variations in responses, missed opportunities for partnerships, insufficient investment in human resources and information gaps on patient outcomes and indicators to measure the performance of healthcare systems [10, 12, 13, 15, 16].

There is also emerging, albeit mixed evidence on the response of low-and middle-income countries (LMICs) to the COVID-19 pandemic. The experience of Kerala state in India during the first wave of infections suggests that a comprehensive multi-sectoral response, combined with social mobilisation, effective prevention measures, community partnerships, learning from previous epidemics, and cushioning the pandemic impact on poor people accounted for successful pandemic management at the time [22]. A 2021 study on Thailand's COVID-19 response found that the country's long-term investment in the health system resulted in an effective response that limited the spread of infections [25]. However, the study found that Thailand's pandemic response has been deficient in addressing the multiple social and economic dimensions of the crisis, engaging with various stakeholders and supporting community networks [25]. Other scholars have highlighted the inadequacy or failures of the COVID-19 pandemic responses in Brazil [26], Iran [19] and Nepal [21], illustrated by increased COVID-19 infections and deaths, poor contact tracing, poor case containment, overwhelmed health care systems, and immense pressure on frontline health workers.

In Africa, initial predictions highlighted the potential devastation of the pandemic in the continent, given its complex disease burden, population vulnerabilities, resource constraints and weak health systems [27], exacerbated by the inequitable distribution of power and resources [28]. Early modelling studies in Nigeria [18] and Kenya [11] have underscored the gaps in hospital capacity for a potential surge in COVID-19 caseloads and the need for additional investments in the health sector, but did not incorporate the perspectives of key health policy actors [11, 18].

South Africa, with an estimated 2021 population of 60.1 million [29], is a constitutional democracy, with three (national, provincial and local) distinctive, yet inter-dependent spheres of government [30]. There are nine provinces, with health a concurrent responsibility of national and provincial government [30]. In practice, the nine provincial health departments function in a semi-autonomous manner, with provincial variations in resource availability, capacity and leadership, resulting in different interpretations and implementation of national laws and policies [31].

The health system is characterised by an inequitable, two-tiered structure: a resource-constrained public health sector that provides care to around 83% of the population, and a large private health sector that only covers 17% of the population [32]. Private health insurance coverage varies both by province and by race. In 2017, around a quarter of people in the urban provinces of Gauteng and the Western Cape had access to private health insurance, compared to less than 10% of people in the rural province of Limpopo [32]. Similarly, 10.1% of black Africans had access to private health insurance, compared to 72.4% of whites [32].

South Africa is recognised globally for its swift and early response to the pandemic [33]. Commencing with a declaration of a national state of disaster on 15 March 2020 [34], the initial response consisted of a strict lockdown, confinement of individuals to their place of residence, except for the production and/or provision of essential services, and a general shutdown of the economy [33]. Notwithstanding the stringent measures, by 26 October 2021, the country had experienced three epidemic waves, a reported cumulative total of around 2.9 million COVID-19 positive cases and 88 934 deaths [1].

Since the declaration of the national state of disaster in South Africa, there is increasing scholarly attention on the COVID-19 pandemic response and management in South Africa [35–42]. Positive aspects include resource mobilisation, centralised and decentralised incident

management teams, rapid evidenced-based decision-making, digital innovation and increased health sector capacity [39, 42]. Since March 2021, the country has embarked on a mass vaccination drive, with more than 20 million vaccine doses administered by 23 October 2021 [1]. However, reported shortcomings or challenges in the country's pandemic response and management include a combination of pre-existing health system weaknesses, leadership failures, corruption, the diversion of financial and human resources, suspension of routine services and deterioration in information collection and management [35–37, 43]. Importantly, the focus of most of the reviews or studies is on the national response, with a dearth of empirical studies that focus on the sub-national level.

Consequently, the aim of this study was to describe the health system response to the COVID-19 pandemic in Gauteng province during the first epidemic wave of infections and to explore the perspectives of key informants on the provincial response. The study was part of a larger case study commissioned by the Office of the Gauteng Premier to document the lessons learned and develop recommendations for the management of future crises.

Notwithstanding the commissioned case study, our rationale for the Gauteng health system study was both its scholarly contribution, and its health policy relevance. A qualitative case study can generate context-specific knowledge on the COVID-19 pandemic response in Gauteng, which is the industrial and commercial powerhouse of South Africa [44]. The case study can also document the key lessons learned and inform the provincial response to future health crises. Hence, our research questions were as follows: What measures were taken to prevent or limit the spread or impact of COVID-19 in Gauteng? What were the perspectives of key policy actors/stakeholders on the implementation of the COVID-19 health system response? What were the health system innovations, strengths, weaknesses and/or unintended consequences of the response in Gauteng? What are the lessons learned from the response during the first COVID-19 pandemic wave? What are possible recommendations to strengthen the Gauteng health system response and to ensure preparedness for future health crises or health system shocks?

## Materials and methods

### Conceptual framework

The stated goals of the COVID-19 pandemic response in Gauteng province are to prevent COVID-19 infections, save lives and ensure a just economic recovery [45]. We combined and adapted the conceptual framework of the InterAction Council on Pandemic Emergency Response to COVID-19 [7] and WHO's COVID-19 Strategic Preparedness and Response (SPRP) Monitoring and Evaluation Framework [46] (Fig 1).

Any conceptual framework is an analytical tool, and the different elements are integrated and cannot be separated in practice. However, we have presented them separately for the sake of clarity.

In our conceptual framework, we consider leadership, management and governance (LMG) as critical to a successful COVID-19 response, and the proverbial roof that anchors the four pillars of the response. In our framework, LMG includes the existence of legislation, policies, strategies, and/or plans, the establishment of COVID-19 emergency structures and/or committees whose role is to ensure oversight, successful implementation and accountability, and intelligence or information without which planning is impossible [47]. LMG also includes coordination of, communication and engagement with various health policy actors, such as health managers, frontline health care providers, and communities.

The health system response rests on four critical pillars: (1) surveillance, containment and control; (2) health service preparedness and treatment (3) resilient health care system; and (4) community engagement and reciprocity.

**Fig 1. Pandemic emergency response conceptual framework.** Sources: Adapted from InterAction Council, 2020 [7] & WHO, 2020 [46].

The first important pillar is surveillance, containment and control. COVID-19 surveillance is critical to prevent a widespread epidemic that could overwhelm the health system, while containment and control measures aim to keep ahead of the epidemic curve and to stop the spread of infection [7]. The activities in this pillar include testing, case finding, contact tracing, hygiene (hand washing and sanitisers), wearing masks, social distancing, isolation and/or quarantine [7, 48].

The second pillar of health service preparedness and treatment aims to prepare the surge capacity of the health sector, and includes triage systems, [re]deployment of staff and appropriate infrastructure e.g. beds, oxygen, ventilators, and personal protective equipment (PPE). This pillar also includes appropriate treatment and care of individuals diagnosed with COVID-19 and requiring hospitalisation [7].

The third pillar of a resilient health care system focuses on maintaining essential health services, engaging and partnering with the health workforce, ensuring supportive and safe

practice environments for health workers, operational support and logistics, and ensuring resource sustainability [7, 46].

The fourth pillar of community engagement and reciprocity aims to prioritise vulnerable populations, and prevent wide-spread transmission, through their participation in the COVID-19 response, encouraging the community to take responsibility for their own health, partnering with civil society, and social mobilisation to ensure information, education and communication on the public health measures to combat COVID-19 [7].

We have used this conceptual framework as an analytical tool to explore the innovations, contestations and fragilities in the COVID-19 response as reflected in our aim. We have also used it to analyse relevant government documents, design the questions in the interviews schedules, and analyse the results of the key informant interviews.

## Study setting

The study setting was Gauteng province, home to approximately 15.8 million people or 26.3% of South Africa's population [29]. In addition to generating 34% of the gross domestic product in 2017 [44], Gauteng has three of South Africa's metropolitan municipalities: the cities of Johannesburg (the country's financial capital), Tshwane (the country's administrative capital), and Ekurhuleni (location of the South Africa's largest international airport), and the two district municipalities of Sedibeng and the West Rand. Hence, Gauteng province is of strategic importance both in South Africa, and in sub-Saharan Africa.

At the time of the study in the last quarter of 2020, Gauteng accounted for 32.0% (n = 225 181) of South Africa's reported COVID-19 cases and had a cumulative incidence risk of 1453.9 cases per 100 000 persons [49].

Between March 2020 and 2 October 2021, the province experienced three epidemic waves [50]. By 2 October 2021, Gauteng reported a cumulative 916 848 COVID-19 cases (31.5% of total), which translates to a cumulative incidence risk of 5 919.7 per 100 000 persons, compared to a national cumulative incidence risk of 4 874.7 per 100 000 [50].

For the period from March 2020 until 2 October 2021, Gauteng had reported a cumulative total of 19 405 COVID-19 deaths [51]. Although there is no universal definition of excess mortality, the South African Medical Research Council (MRC) has analysed the number and rates of COVID-19 excess deaths in South Africa by province and by municipality [52]. For the same period until 2 October 2021, the MRC has reported 55 311 excess deaths in Gauteng [52]. This translates to an age-standardised excess death rate of 390 per 100 000 for Gauteng, lower than the rate of 442 per 100 000 for South Africa as a whole, but higher than the rate of 341 per 100 000 for the Western Cape province [52].

## Study design

This was a qualitative case study design, using mixed methods that consisted of an analysis of government documents and in-depth interviews with key informants.

## Study period

The focus of the case study was on the health system response in Gauteng during the first COVID-19 epidemic wave, namely from March 2020 until September 2020.

## Document analysis

The purpose of the document analysis was twofold. Firstly, to examine the phrasing and content of COVID-19 policies and/or progress reports and the evolution of the pandemic response

in Gauteng. Secondly, to allow for triangulation of various sources of data. We focused on the period from 1 March 2020 (roughly the start of the pandemic in South Africa) until September 2020, which roughly corresponds with the end of the first wave of COVID-19 infections.

We used the READ (Ready materials, Extract data, Analyse data, Distil) approach, which is a step-by-step guide to analysing documents in health policy research [53]. The first step was to find or "ready" our materials for review. We did this by requesting and searching for official national, provincial and local government documents pertaining to COVID-19 response in general, and the health system response in particular.

We identified 17 documents of relevance to our study period. These included national, provincial and local government documents. The national documents included the National Disaster Management Act and accompanying regulations, health regulations, surveillance reports and media updates and/or press releases. The provincial documents included the COVID-19 strategic plan, reports or presentations to the Provincial Cabinet or the Provincial Coronavirus Command Council, surveillance reports, and media updates or press releases. Similarly, we searched for or requested specific local government COVID-19 policies or reports.

In the next step, the principal researcher extracted the relevant information from the 17 documents and entered the data into a grid. Our adapted conceptual framework (Fig 1) and our research aim guided this process of extraction and subsequent inductive analysis. The distilling (i.e. findings) of the document review are integrated with those of the key informant interviews.

## Key informant interviews

**Participant selection.** We used purposive sampling to select key informants based on their knowledge and/or experience of the COVID-19 health system response in Gauteng, and the implementation of the overall strategy in the five health districts and the various health facilities. We selected individuals from the following six categories of key informants: executive or senior managers or officials from different levels of provincial or local government; senior clinicians or frontline health workers in hospitals and clinics; professional associations and/or unions; academics, researchers or technical experts; the private health sector; civil society/ non-governmental organisations or health advocacy groups.

**Data collection tool.** We developed two interview schedules in English: one for executive managers and one for all the other key informants (S1 Appendix). The semi-structured interview schedule for executive managers consisted of five main subsections. The first section focused on the key informant's role in the COVID-19 response. The second section focused on a description or overview of the actual health system response to COVID-19, and included probes related to LMG and the first three pillars in the conceptual framework. The third section focused on stakeholder involvement and communication, and included probes on the fourth pillar of the conceptual framework. The fourth section focused on key informants' perspectives on the COVID-19 response and/or strategy, notably their perspectives on the innovations, contestations or weaknesses, unintended consequences (both positive and negative) of the COVID-19 response and its implementation. The last section focused on key lessons from the management of COVID-19 interventions in Gauteng and recommendations for future management of health crises (including a possible second wave).

The semi-structured interview schedule for all other key informants was a truncated version of the key informant schedule for executive managers, and excluded sections 2 (specific response) and 3 (stakeholder involvement).

The research team reviewed the interview schedules for content validity and clarity of questions.

**Research team and preparation for data collection.** The research team consisted of four individuals. The principal researcher (LCR) holds a South African Research Chair, has a PhD in health systems research, and has more than three decades of health systems research, leadership and management experience. The second author (CM) is a retired public health medicine specialist and paediatrician, with four decades of management and health system experience. The third author (BM) is a PhD candidate, holds both an undergraduate health professional qualification and a Master of Public Health degree. The final author (SM) holds an undergraduate health professional qualification, a Master's degree in Nursing Education and a PhD in health systems research. Cumulatively, BM and SM have two decades' experience of working as health professionals.

The research team jointly developed and discussed the interview schedules prior to their finalisation. We also held a separate meeting to go through the questions and did two trial runs to ensure readiness to interview the key informants and to collect the data.

**Data collection.** We conducted the interviews between 1 September and 15 October 2020. Each participant was contacted via email to request voluntary participation in the study. Following informed consent, we arranged the interview date and time with each key informant. The researchers emailed the information sheet, informed consent forms, and the relevant interview schedule to the study participants. All interviews were conducted in English and virtually via Microsoft Teams or Zoom. We conducted telephone interviews on two occasions when technology failed.

Each interview began with an introduction of researchers to the participant, assisting the participant to be familiar with the virtual platform and putting the participant at ease. This was followed by an introduction to the study, and an explanation of the voluntary nature of participation. Prior to the start of the interview, the researchers confirmed voluntary consent for participation and recording of the interview.

Following informed consent, the research team used the semi-structured interview schedule as a guide to explore each participant's perspectives on the COVID-19 response and the management of the COVID-19 pandemic in Gauteng. The researchers used probes to obtain details and clarification of responses. Each interview lasted between 30 to 60 minutes, but the duration varied depending on the key informant's responses. We made detailed notes during the interviews, and complemented these notes with the audio-recordings.

All interviews were recorded digitally and labelled with a key informant code. All audio-recordings are kept on a password-protected computer to ensure confidentiality.

**Data analysis.** Following data collection, the principal researcher examined and collated all the raw data and submitted the collated data to the other researchers for review. The four researchers held several meetings to discuss progress, debrief on the completed interviews, identify the emerging issues, identify gaps and assess whether new information emerged from the interviews.

Each of the four researchers analysed the data independently, using thematic analysis [54]. Each researcher read and reread each transcript independently to familiarise herself with the data, and to get a sense of the whole interview. The transcripts were coded line by line, by writing key words on early impressions, using the direct words from the transcripts [54]. Each researcher made notes on reflections from the data. Following the coding stage, each researcher developed themes, interrogated and evaluated the themes for similarities and differences in meaning across different categories of key informants.

Once the coding was complete, the four researchers held a meeting to discuss the independent codes and themes, and to reach inter-coder agreement. Following this, the research team held several other meetings to analyse the data, discuss and finalise the themes, their meaning and interpretation. We used an iterative process of inductive coding and theme development,

followed by interrogation of these codes and themes in light of the study objectives and the selected conceptual framework.

**Ethical considerations.** The Office of the Gauteng Premier commissioned the original case study. All researchers signed confidentiality and non-disclosure agreements. The principal researcher (LCR) requested advice from the Chair of the Human Research Ethics Committee (Medical) of the University of the Witwatersrand in Johannesburg on whether an ethics submission should be submitted prior to conducting the interviews. The Chair indicated that the confidentiality and non-disclosure agreements were sufficient to undertake the Gauteng case study.

During the data collection, the research team provided a detailed information sheet to all key informants, as well as consent forms for conducting the interview and for the audio-recording. Prior to the start of the interview, we provided a verbal explanation of the study, and informed participants of the voluntary, confidential, and anonymous nature of study participation.

The researchers obtained written approval from the Office of the Premier to submit this journal article based on the health case study for possible publication. Subsequently, the principal researcher also obtained an ethics waiver from the Human Research Ethics Committee of the University of the Witwatersrand in Johannesburg (#W-CBP-210331-03) to enable publication.

We complied with the principles of the Singapore Statement on Research Integrity [55], and use participant codes to ensure anonymity. All the data are stored on a password-protected laptop. The audio-recordings will be destroyed two years after publication of the case study.

**Trustworthiness and validation.** The researchers ensured credibility by prolonged engagement with the documents, the key informant data and the audio recording of the interviews [56]. We ensured dependability by having a semi-structured interview schedule and a detailed description of the methods, data collection and analysis. Conformability was ensured through independent coding of the key informant interviews, discussion of, and agreement on the themes by the four researchers [56, 57]. We validated the data by doing two presentations to key stakeholders in the Office of the Premier, and by submitting a detailed written report.

As the interviews were conducted after the first wave of COVID-19 infections, and in light of a second wave of infections between December 2020 and 15 February 2021, we sent the interview transcript and an abstract of the case study to 15 key informants in February 2021. We requested these individuals to comment voluntarily on their original interview responses, and the summary of the results. Six of the key informants responded, confirming their initial responses, and support of the emerging themes. One key informant added more details to her original transcript, but without altering the core messages obtained during the first interview.

## Data triangulation and integration

In the final step of the data analysis, the research team triangulated the information and generated the final themes through a process of immersion, reflection and discussion [58]. This process combined our conceptual framework, the study objectives, document analysis and the themes from key informant interviews.

## Results

### Characteristics of key informants

We interviewed 36 key informants (KIs), and their characteristics are shown in Table 1.

**Table 1. Characteristics of key informants interviewed.**

| Category of key informant | Male | Female | Total |
|---|---|---|---|
| Executive manager in provincial or local government | 5 | 2 | 7 |
| Facility/ district managers, senior clinicians or frontline health workers in hospitals and clinics | 6 | 5 | 11 |
| Professional associations and/or unions | 2 | - | 2 |
| Technical support, research, academia | 3 | 5 | 8 |
| Private health sector | 1 | 2 | 3 |
| Civil society | 4 | 1 | 5 |
| **Total** | **21** | **15** | **36** |

## Emerging themes

Five inter-related themes emerged from the document analysis and the key informant interviews: (1) innovative, multi-sectoral provincial response to COVID-19; (2) contestations and contradictions in the pandemic response; (3) fragilities, complexity and vulnerabilities of existing systems; (4) under-investment in the health workforce; and (5) various missed opportunities. Table 2 shows the themes and sub-themes, and their alignment to the selected conceptual framework. Although the themes overlap, each is described separately for the sake of clarity. Within each theme, we present the findings from the document analysis and the interview themes in an integrated manner.

**Theme 1: Innovative, multi-sectoral and comprehensive provincial COVID-19 response.** The first theme, derived from both the document review and the interviews, centres on the multi-sectoral and comprehensive response to the COVID-19 pandemic, and the innovations, whether driven from the top, or those that happened organically.

The National Disaster Management Act provides the legislative framework for the COVID-19 pandemic response in South Africa [34]. In concert with national developments, the Gauteng Provincial Government adopted a strategic response to the COVID-19 pandemic that aimed to address the dual challenge of saving lives and saving the economy [45]. The Gauteng COVID-19 Strategic Response incorporated a detailed risk assessment that took account of multiple levels of deprivation at the district level and consisted of six pillars: a comprehensive health response; food security and social relief; state capacity and adaptability; economic response; social mobilisation and human solidarity; and law enforcement and compliance [45].

Several high level governance structures were set up, the most important of which were the Provincial Coronavirus Command Council (chaired by the Premier) and the Provincial Disaster Management Command Centre (PDMCC) co-chaired by the Provincial Director-General and the Provincial Police Commissioner [45]. Each of the six pillars translated into a work stream that presented updates to the PMDCC, which in turn reported to the Provincial Coronavirus Command Council, and to the Provincial Executive Council (i.e. the Provincial Cabinet).

Several key informants commented positively on the visible and strong political leadership by the Gauteng Premier and the former Member of the Executive Council (MEC) for Health.

*We were led properly in our response. The Provincial Command Centre had given us clear, accurate instructions on what needs to be done e.g. the setting up of quarantine centres. It showed bold leadership* (KI 8, Central Hospital Manager).

Two key informants from civil society organisations also echoed the strong, visible political leadership. One said:

**Table 2. Emerging themes, sub-themes and links to the conceptual framework.**

| Theme | Sub-themes | Links to conceptual framework |
|---|---|---|
| Innovative, multi-sectoral and comprehensive provincial COVID-19 response | • Inter-departmental governance structures<br>• High-level and visible political leadership<br>• Supportive hospital and district managers encouraged teamwork and local-level innovation<br>• Responsiveness and agency of frontline staff<br>• Additional financial resources<br>• Central warehousing and dedicated budget for personal protective equipment (PPE)<br>• Digital innovation and partnerships<br>• Institutionalisation of public health measures | • Roof: Leadership, management and governance<br>• Pillar 1: Surveillance, containment and control<br>• Pillar 2: Health service preparedness and treatment<br>• Pillar 3: Resilient health care system<br>• Pillar 4: Community engagement and reciprocity |
| Contestations and contradictions in the pandemic response | • Top-down, centralised approach<br>• Intergovernmental relations<br>• Appropriateness of regulations and guidelines<br>• Differential involvement of stakeholders in strategy development and/or decision-making<br>• Quality of information<br>• Results of risk modelling<br>• Decision on field hospitals<br>• Utilisation of isolation or quarantine facilities<br>• Transparency and accountability<br>• Suitability of communication strategies | • Roof: Leadership, management and governance<br>• Pillar 2: Health service preparedness and treatment<br>• Pillar 3: Resilient health care system |
| Fragilities, complexity and vulnerabilities of systems | • PPE corruption<br>• Single-minded focus on COVID-19<br>• Health system collateral damage<br>• Lack of clarity on testing or screening guidelines<br>• Laboratory testing capacity<br>• Sub-optimal health information systems<br>• Contradictory or fearful messages to communities | • Roof: Leadership, management and governance<br>• Pillar 1: Surveillance, containment and control<br>• Pillar 2: Health service preparedness and treatment<br>• Pillar 3: Resilient health care system<br>• Pillar 4: Community engagement and reciprocity |
| Under-investment in the health workforce | • Insufficient involvement of frontline staff<br>• Variations in human resource management<br>• Failure to address the psycho-social needs of health workers<br>• Inadequate orientation and delays in staff training in infection prevention and control<br>• Insufficient or lack of PPE | • Roof: Leadership, management and governance<br>• Pillar 2: Health service preparedness and treatment<br>• Pillar 3: Resilient health care system |
| Missed opportunities | • Primary health care/ community based response vs hospi-centric, medicalised response<br>• Engagement with communities and/or civil society<br>• Enhanced regulation and/or partnerships with private health sector | • Roof: Leadership, management and governance<br>• Pillar 3: Resilient health care system<br>• Pillar 4: Community engagement and reciprocity |

*The Premier has given the impression of efficiency and transparency, and that has inspired confidence, and the belief that everything was under control. I was impressed with the slides presented at the media briefings, and the attempt at showing hard data* (KI 23, Civil Society).

Several key informants reported that the pandemic provided an opportunity for innovations at various levels of the health system. In one health district, one key informant reported that contact tracing started early March 2020 when the country was in hard lockdown, with very few reported COVID-19 cases. The district health team developed written standard operating procedures. They established two types of tracer teams: physical tracers (primarily nurses) whose responsibility was to visit the homes of COVID-19 cases to test and educate other members of the households, and telephonic tracers (lay counsellors or other tracers), who were responsible for monitoring the cases and contacts every day for the 14-day isolation period.

The public sector hospital managers and clinicians interviewed were of the opinion that the success of the COVID-19 response in Gauteng was largely because of proactive managers at health facilities, the excellent teamwork, innovation or improvisation necessitated by the pandemic, and the contribution and sacrifice of frontline staff.

*The response from the hospital was quite quick. By January* [2020], *we already started training on COVID-19. The team work was amazing–everyone pulled together to get the work done–they were paid overtime, the hospital hired agency nurses -the province only gave extra staff from 1 June 2020. Our psychiatrist set up an online psychosocial support for staff, which was then taken up by province. We did all the protocols- we gave them to province, not the other way.* (KI 7, Central Hospital Manager).

Key informants reported that the pandemic highlighted the thousands of committed and responsive health workers, especially those on the frontline, who rose to the challenge of providing health care, in line with their ethical and professional obligations.

*Clinicians from different departments organised themselves to work as a team and to cover departments that needed the most assistance. Psychologists provided debriefing for staff members* (KI 8, Central Hospital Manager).

*COVID-19 brought out the strength of each individual health worker. We have never experienced better teamwork than during this period. . . there was willingness to share resources and to work overtime* (KI 9, Regional Hospital Manager).

In Gauteng, there was recognition of the need to ensure financial, human and other resources to meet the increased demand for hospitalisation and quarantine. The Gauteng Provincial Government allocated an additional R4 billion (~267 million US$; $1 = R15) to the health sector for the appointment of additional staff, the procurement of PPE, and the upgrading or building of additional infrastructure. Many key informants viewed this significant injection of finances in a positive light.

*The response to COVID-19 was comprehensive. We were able to get additional finances. We also took the decision to revamp the current infrastructure, using alternative building technology (*KI 4, Executive Manager).

*The support was palpable on the ground. . .we were given R14 million (~$933 000) to beef up current staffing levels. The budget mitigated staff shortages especially with the opening of new wards (*KI 8, Central Hospital Manager).

Key informants from the frontline expressed their appreciation for the innovative initiative to have a central warehouse and dedicated budget for PPE.

*The PPE procurement was centralised, we fetched our PPE from the warehouse-the system was working well for us* (KI 8, Central Hospital Manager).

*Initially we had challenges* [with PPE] *but later, the district did their best to get us enough PPE, hence we had limited number of staff who were infected because we had enough PPE in my facility* (KI 32, PHC Facility Manager).

Some of the key informants commented on the digital innovation in Gauteng, evidenced by the screening/tracing app, and the bed availability dashboard. Gauteng province also

sponsored a consortium of modelling experts to identify potential hotspots, and to estimate the need for hospitalisation, critical care beds and resources.

Although public health measures such as handwashing, sanitisers, and mask wearing were national interventions, several key informants highlighted the widespread positive behavioural change and institutionalisation of these measures in the province.

**Theme 2: Contestations and contradictions in the pandemic response.** The second emerging theme highlights key informants' perspectives on the contestations and/or contradictions in the COVID-19 response in Gauteng.

Key informants from the frontline complained about a top-down approach to planning, with instructions issued, lack of or insufficient engagement with hospital managers, and little or no room for questioning decisions from the top. Those in the private health sector commented on the lack of engagement or discussion on contracts, with perceptions of rigidity and a heavy-handed, top down approach of the Gauteng Department of Health. In some instances, the highly centralised approach resulted in confusion or duplication of roles.

*At the War Room* [the centralised structure set up to coordinate and guide the COVID-19 response], *there was duplication of roles, you would find three sets of teams doing the same thing. You would have a* [provincial] *public health specialist team analysing data, then you have a clinical health team in the department (of health) doing the same thing, and then the district people (health programmes) doing the same thing (*KI 9, Regional Hospital Manager).

However, some informants were of the view that the province was at the mercy of instructions issued by the National Department of Health.

*National including NICD* [National Institute for Communicable Diseases], *province, district– it was very unclear who did what. The War Room was good but it wasn't clear at first what kinds of decision were taken there. This led to a certain paralysis in responding. For example, in early March 2020, we were contacted by a church group planning a big event in April, asking for advice. Province said they couldn't advise, the NICD said they couldn't–eventually the District had to give an answer. We emphasised that we had no authority to do so, but said that it would not be a good thing to go ahead with the event (*KI 12, District Health Manager).

There were mixed perceptions on intergovernmental relations, and coordination and collaboration. Some key informants were of the opinion that COVID-19 facilitated improved collaboration across several levels, namely across provincial government departments, between the province and local government health departments, between the public and private health sectors, between tertiary hospitals and the surrounding clinics, and within hospitals, between management and staff.

In contrast, some key informants highlighted poor intergovernmental relations. One key informant reported that one of the municipalities only came on board in August 2020, while another said:

*Intergovernmental relations were problematic-although the City owns clinics, the provincial chief director communicated directly with the clinics, and bypassed the executive director for health in the city. There was insufficient investment in good relationships, and at times, the province-local government politics played itself out (*KI 1, Executive Manager).

Another key informant highlighted the fragmentation between national, provincial and local government, lack of or insufficient communication, and weak messaging.

*The left hand does not know what the right hand was doing. There was a huge disjuncture between National and the Gauteng Department of Health, lack of trust, and the relationship has worsened. The national meetings were mostly tick box exercises* (KI 6, Technical Support/ Researcher/ Academic).

All the key informants were familiar with the rationale and purpose of the Disaster Management Act, namely to ensure an integrated and coordinated response to the COVID-19 pandemic. Some highlighted the confusion created by the accompanying regulations, especially at the beginning of the lockdown, exacerbated by the numerous changes, the lack of a coordinating mechanism, and lack of user-friendly guidelines. Those key informants from the frontline complained about the rapidly changing regulations that were open to interpretation, and the impracticality of some of these regulations.

*There were regulations about the numbers of people that have to be at work. In health, all employees are essential and we couldn't keep to the 30% required by the regulations. The regulations stated that people above 60 and with co-morbidities, were supposed to stay at home. In health, these were the people with the wisdom to deal with certain complications that might arise due to COVID-19, such as highly skilled critical care nurses* (KI 10, Central Hospital Manager).

Another key informant highlighted the problem of unclear guidelines.

*We faced challenges with unclear guidelines. For example, when a facility has a COVID-19 case, it is not clear who should do the deep cleansing—the general assistants at the facility or the external service provider* (KI 32, PHC Facility Manager).

Two private sector key informant also highlighted the difficulties of unclear guidelines or rapidly changing regulations.

*The regulations were amended every 14 days, and it was problematic to adapt so quickly* (KI 5, Private Sector Manager).

*The guidelines were not for the entire nation, and were not applicable to the private health sector. There was no opportunity for private sector input. We reached out to National and assisted with the wording of the regulations for the prescribed minimum benefits, to make sure that people are covered for COVID-19* (KI 24, Private Sector Manager).

Although the provincial COVID-19 Strategic Response emphasised the involvement of different stakeholders in fighting the pandemic [45], some key informants criticised the lack of participation of communities, frontline health workers, and the private health sector in decision-making.

Notwithstanding the innovative provincial initiative on modelling COVID-19 cases and resource requirements [45], the key informant interviews revealed intense contestations on the quality of information, and the results of the modelling, especially the estimated number of critical care and general hospital beds.

*There was panic at the beginning, with over-estimates of the actual number of cases. For example, the 23 000 active cases look very different to projections of more than 100 000 cases. The additional* [financial] *allocation to health was made following their modelling and projections. The* [Gauteng] *Department of Health initially requested an additional R14.9 billion*

*(~ \$1 billion) for PPE, oxygen and infrastructure. . .we allocated R4 billion* (KI 3, Executive Manager).

The initial 2020 modelling exercise produced an over-estimate of the expected number of COVID-19 cases at the peak of the pandemic in Gauteng [45]. This in turn led to an over-estimate of critical care and general hospital beds in the province, which influenced subsequent decisions or strategies that aimed to address the gap between bed needs and bed availability. Although the provincial government included the private health sector in the original modelling, the subsequent planning excluded the bed availability in the private health sector [45]. Consequently, the provincial government planned to overcome the gap in the need for additional beds through repurposing of existing wards, decanting wards in existing facilities and preparing them for the use of COVID patients, creating additional capacity within the existing hospital platforms, and creating temporary capacity outside existing facilities through field hospitals [45].

Key informant interviews also revealed contestation on the decision-making and actual decision on field hospitals. Some key informants were of the opinion that the decision on field hospitals was taken by the national Ministers of Public Works (Infrastructure Development) and Health, and handed down to Gauteng province, with no consultation or input from anyone. However, the research team was informed that Gauteng decided to invest in its own public health infrastructure, in line with the 2030 Growing Gauteng Together plan of action. Hence, only one field hospital was erected in the South of Johannesburg.

Some of the key informants thought that the field hospitals in preparation for the surge demonstrated quick decision-making in response to the pandemic, and created additional needed capacity.

*We were able to develop additional capacity, using alternative building technology, which will remain the legacy of the pandemic. These facilities will be put to alternative use, e.g. for medical oncology* (KI 21, Executive Manager).

Others were of the opinion that there was lack of transparency and that the decision on field hospitals was both controversial and ill-advised.

*Field hospitals were very controversial from the start, as the experience from many places shows that they were not used. They are very badly designed and criticism or requests for change were ignored. Lots of money went into these. The decision-making was not agile enough; it was too compartmentalised, and unable to change direction* (KI 11, Central Hospital Clinician).

*There was no information in the public domain about field hospitals, whether they were used to their capacity, how other hospitals were coping and managing COVID-19 numbers* (KI 13, Civil Society).

The key informants also expressed mixed opinions on isolation or quarantine facilities. Some key informants were of the opinion that the setting up of the quarantine centres showed bold provincial leadership. Others pointed out that there was poor utilisation of quarantine facilities, because the amenities were basic and/or community members refused to leave their homes. This was exacerbated by the long period of quarantine (14 days) in the early lockdown stages.

Lastly, the interviews revealed contradictory perspectives on the suitability of communication strategies used in the province. Many key informants pointed to good high-level

communication from the Premier and former MEC on COVID-19 as a public health crisis, and government prioritisation of the pandemic. There were regular updates on the strategy and COVID-19 updates in the mainstream media (television, radio, newspapers). They felt that the explicit support of national decisions by the Premier and MEC was critical in getting a unified response. The visits of politicians to health facilities and communication with staff also strengthened Gauteng's response to COVID-19.

However, key informants from civil society were of the opinion that the Premier adopted the same model of communication as the President, and that this was inappropriate in Gauteng, as the small size of the province provides a platform for greater interaction with various stakeholders. Although there was recognition of the Premier's leadership, individuals from civil society organisations lamented government's non-responsiveness to their offers of support or collaboration.

Some senior government officials, individuals from civil society and the private health sector highlighted sub-optimal communication to communities.

*The lack of involvement or briefing of SALGA (= South African Local Government Association), trade unions, SANCO (= South African National Civics Organisation)-I think it was a big error at provincial level. The communication with the community was very, very, very poor (*KI 1, Executive Manager).

*The communication was very sophisticated, but it did not filter to community level* (KI 23, Civil Society).

*I think the provincial government should use existing community forums to communicate key aspects of the pandemic, adopt a non-prescriptive approach and use the existing capacity to the maximum (*KI 24, Private Sector Manager).

**Theme 3: Fragilities, complexity and vulnerabilities of systems.** The document analysis revealed detailed plans on epidemiology and surveillance, case management, health infrastructure, laboratory services, emergency medical response and research [45]. The June 2020 report on PPE included statistics on the different types of PPE, existing stock, as well as stock ordered. Although the central warehousing and procurement of PPE were a Gauteng innovation, during the second half of 2020, widespread reports surfaced on allegations of corruption in the procurement of PPE [59, 60]. The subsequent referral of the PPE corruption to the Special Investigating Unit illustrated that greed appeared to trump ethical conduct and accountability [59].

Many key informants highlighted the reported PPE corruption scandal as a painful experience, which overshadowed the achievements and strengths of the COVID-19 response in the province. They were of the opinion that the alleged corruption was entirely preventable, and that the weaknesses of existing systems, with lack of checks and balances, created the perfect opportunity for corruption. They pointed out that a combination of centralised procurement, lack of transparency, lack of accountability, failure to prioritise societal interest above self-interest, non-compliance with existing legislation, and a failure to identify and anticipate risks contributed to the alleged corruption.

Several key informants pointed out that the COVID-19 pandemic exposed the fragilities, and in some instances, the failures of the health system. These fragilities were exacerbated by the single-minded focus on COVID-19, with its prioritisation resulting in a *de facto* health system lockdown, especially of the public health sector. This had many unintended negative consequences, including a reduction in access and/or provision of essential health services (e.g. maternal and child health services, HIV and tuberculosis, non-communicable disease care),

scaled-down hospital services (e.g. no elective surgery), and loss to follow-up of some patients. Some key informants pointed out that the impact of COVID-19 prioritisation on adverse events e.g. maternal and/neonatal deaths is unknown.

*Collateral damage to patient care was and is significant. The deaths of non-COVID patients probably surpassed those of COVID patients for the first 3 months although no formal system was set up to measure this. Many patients died without proper care while waiting for their test results* (KI 11, Central Hospital Clinician).

*There has been a 25–50% reduction in* [childhood] v*accinations–this is recovering but there is a large unvaccinated cohort and we run the risk of a massive measles outbreak with increased mobility. There were major disruptions to HIV and tuberculosis treatment. We raised concerns about collateral damage very early on* (KI 15, Technical Support/ Researcher/ Academic).

Some key informants pointed to the collateral damage in terms of the quality of care provided.

*The quality of care was not the best—we had to erect tents. In the beginning, medications were not administered as needed because of limited personnel* (KI 10, Central Hospital Manager).

The quality of care was also affected by the appointment of junior staff and/or the reliance on agency nurses.

*Our effort to circumvent overcrowding made us to send the newly appointed staff to casualty. This was a serious problem—you find that there were 15–20 patients with three or four nurses who didn't understand the functional mandate of a particular department. We discovered many clinical service gaps* (KI 9, Regional Hospital Manager).

Several key informants were of the opinion that the perceived collateral damage was exacerbated by fearful and apparent contradictory messages to communities.

The document analysis revealed the establishment of a separate Public Health Stream within the health department that presented detailed reports to the Provincial Advisory Council, which in turn reported to the overall governance structure, namely the Provincial Coronavirus Command Council [45]. On paper, there were detailed strategies and procedures that dealt with epidemiology and surveillance, community mobilisation, education and advocacy, and case management and public health training [45].

However, key informant interviews demonstrated a lack of clarity or knowledge on the provincial COVID-19 testing strategy.

*There was failure to share the Provincial Testing Strategy, especially on the employee testing. There was also non-compliance with testing guidelines e.g. random testing of communities without following the guidelines, leading to conflict with our laboratory team* (KI 20, Technical Support/ Research/ Academic).

Frontline staff highlighted the challenges experienced with laboratory testing.

*While we were managing the pandemic, we had a lot of areas that were not in sync, so this is an area we can improve e.g. the laboratory tests, I don't know where the problem was, but we*

*had a lot of instances where people would be reported as negative or positive, but there would be missing information—no addresses and contact numbers, so it made contact tracing diffi- cult* (KI 34, District Health Manager).

Both the document analysis [61] and key informant interviews pointed to the vulnerabilities of health information systems. Key informants highlighted staff shortages, lack of capacity to analyse and/or interpret epidemiological trends, initial reluctance to participate in the national hospital surveillance system, and the fragmentation of the existing information systems, some of which remained paper-based. They pointed to poor data quality, which meant that some- times the information provided did not make sense. The consequences of poor health informa- tion systems were the inability to plan pro-actively, use evidence to inform decision-making, respond quickly, and/or to implement appropriate action. These problems necessitated the set- ting up of new or parallel systems, creating a disjuncture between the data from the NICD and the provincial health information system, duplication of efforts, and lack of reliable informa- tion to strengthen the pandemic response.

**Theme 4: Under-investment in the health workforce.** Although many key informants expressed appreciation for the additional funding and staff appointments as part of the COVID-19 response, a recurring theme in the interviews was the perceived under-investment and insufficient focus on the health workforce, especially on frontline staff. Some key infor- mants were of the opinion that there was lack of prioritisation of human resources and that the health department did not demonstrate appreciation of staff, despite health workers risking their lives to do their duties.

*Doctors are burnt out, nothing you do really is recognised, morale is low, and there is poor support, lack of equipment, unfilled trolleys, staff shortages, having to run around for stock . . .it a very draining emotional experience* (KI 36, Central Hospital Clinician).

*Health care workers contracted COVID 19 and some lost their lives. We should have given a little more attention to frontline health workers e.g. buses provided for travel so that they don't have to contract COVID, more emotional (psychological) support (*KI 32, PHC Facility Manager).

They lamented insufficient involvement of frontline staff in the pandemic response, or in certain decisions, such as putting up separate tents for COVID-19 positive patients.

*We should have been involved in discussions around how oxygen will be supplied to the tent, and that is a problem. . ..sometimes you run out of oxygen during a resuscitation* (KI 36, Cen- tral Hospital Clinician).

Poor or sub-optimal human resource management exacerbated the perceived under-invest- ment in the health workforce.

*COVID-19 exposed poor managers who often took out their frustration on ordinary health workers. This was in contrast to those managers who were able to respond to pressure* (KI 22, Professional Association).

Some of the key informants commented on the proactive response in Gauteng to deal with the serious threat of the COVID-19 pandemic, reflected in a dedicated focus on infection pre- vention and control (IPC) and occupational health services. Although the COVID-19 pan- demic underscored the importance of IPC principles (e.g. handwashing, IPC training and

infrastructure), which have been neglected historically, several key informants pointed out that the large number of infections among health workers reflected weaknesses in IPC.

*We did not prepare the health care workers enough; most of the infections were on the so-called green side* [non-COVID wards]. *In accident and emergency units, there were infections caused by insufficient or poor quality PPEs, but utilisation* [of PPE] *was also a challenge* (KI 4, Executive Manager).

*Central hospital ABC was actually overwhelmed, and ran out of space for patients. There were mini-outbreaks happening in the wards. The infection rates among health care providers are an indictment on preparedness and PPE provision. For example, 55% of health workers in the Internal Medicine wards housing COVID or persons under investigation became infected over a period of 6 weeks. This happened due to poor IPC practices, but also poor PPE* (KI 15, Technical Support/ Researcher/ Academic).

The comments from key informants on COVID-19 infections among health workers were borne out by an August 2020 Ministry of Health media statement [62]. By 4 August 2020, there were 27 360 COVID-19 cases among health workers, the majority (78%) were from the public health sector, while 22% were from the private sector [62]. At the time, the overall infection rate amongst health workers was 5% [62]. The media statement also reported 240 COVID-19 deaths among health workers, the majority (84.6% or 203) from the public health sector [62].

Several key informants pointed out that the provincial health department failed to deal with the fears and anxieties of frontline health workers.

*We thought that we are strong, but we were all paralysed with fear in spite of infection prevention and control training. In June/July 2020 when the surge was really high, even though we had a guideline on which protective PPE should be worn, everyone wanted to wear the triple layer (disposable scrub suit, gowns, coverall, plastic aprons and N95 masks) of PPE even in non-COVID wards because of fear and not because of scientific evidence* (KI 8, Central Hospital Manager).

*This issue of staff fear wasn't recognised- this was an unintended consequence of the communication campaign to try and get the population to follow guidance but had the effect of creating fear and panic amongst staff that this* [COVID-19] *was like Ebola- that they would die. The perception that there was a high mortality associated with* [COVID-19] *infection was never undone, even when reality was showing something different* (KI 11, Central Hospital Clinician).

The majority of key informants in both the public and private health sectors highlighted that there was no clear strategy for employee assistance or psychological support. They noted that there was insufficient acknowledgement of the fears and anxieties of frontline health workers. A senior executive in the private health sector noted the following:

*My sense is that staffing was a kneejerk reaction, rather than planned. Staffing could have been managed better, including sharing of human resources between the public and private health sectors. The two sectors operated separately. At our hospital group, we realised later on in the pandemic that the health workforce is an absolute priority* (KI 5, Private Sector Manager).

Another private sector key informant noted that the early preparation of staff should have included psychological support, as there was a lot of uncertainty and the impact of COVID-19 on staff was massive.

**Theme 5: Missed opportunities.**   The interviews with key informants revealed that there were various missed or wasted opportunities. The Comprehensive Health Response was a hospital-based strategy primarily, with the Ward-Based Battle Plan developed much later, seemingly after COVID-19 infections had peaked in the province. Some key informants criticized the hospi-centric pandemic response rather than primary health care and/or community-based response.

> *We need to move away from a curative system and invest in primary health care (PHC). You cannot talk about PHC and continue to invest in hospitals. COVID-19 was a missed opportunity to optimise and focus on PHC* (KI 18, Civil Society).

There were also perceptions of missed opportunities to engage and partner with communities and/or civil society. Several key informants pointed out that the COVID-19 response did not seem to take into account the vulnerability of communities in informal settlements, who did not have clean running water and adequate sanitation. They were of the opinion that the hard lockdown regulations of social distancing, confinement to residences and hand hygiene in informal settlements were almost impossible to adhere to. Although there was recognition of the Premier's leadership, individuals from civil society organisations lamented government's non-responsiveness to their offers of support, or collaboration.

> *There was a lot of goodwill on the part of civil society, but it does not have the power or the resources to intervene. Government did not take advantage of civil society or capitalise on the experience of extensive community involvement in combatting the HIV pandemic. To a large extent, the response has been biomedical and bureaucratic* (KI 23, Civil Society).

Some of the key informants highlighted the missed opportunity of government in regulating the private health sector to enhance equity, and to ensure access to health care and to scarce resources, such as ventilators.

> *We had a two-sector response to the pandemic, contributing to inequity. There was a different turn-around time based on where you tested e.g. in private sector quick access, but this was not the case in the public sector. We did not have regulations on the distribution of resources e.g. ventilators between the public and private sectors* (KI 18, Civil Society).

These key informant perspectives on the missed opportunity to strive for greater equity between the private and public health sectors were supported by the document analysis [41, 49]. Although the private health sector provides care to less than 20% of the South African population, it accounted for the majority of COVID-19 tests during the first wave of infections [49]. Evidence suggests that there is a lower threshold for admission to private hospitals, thus exacerbating inequities in access to COVID-19 treatment and care [41].

In contrast, those key informants from the private sector felt that they had been excluded, and this was both frustrating, and a missed opportunity for collaboration.

> *The biggest disappointment is that they* [public health sector) *thought they could do it themselves. There was a major missed opportunity to engage the private health sector in service*

*provision. The guidelines were not for the entire nation, and were not applicable to the private health sector* (KI 24, Private Sector Manager)

*I have tried through three parallel pathways to get the private sector involved. . .it has been difficult to know how to assist. There has been no engagement or discussion* [in Gauteng] *on contracts with private medical practitioners (*KI 26, Professional Association).

## Discussion

This case study set out to explore the health system response to COVID-19 in the Gauteng province of South Africa, during the first epidemic wave of COVID-19 infections. At the time of writing in October 2021, Gauteng, as in the rest of South Africa, had experienced three epidemic waves [50]. Notwithstanding the second epidemic wave that succeeded our interviews, six of the key informants validated their original responses and the relevance of the emerging themes.

Both the document analysis and the key informant interviews revealed that the Gauteng province developed an impressive, comprehensive and multi-sectoral COVID-19 Strategic Response, with high-level, visible leadership, and oversight provided by inter-departmental governance structures. Health was one of the key pillars of the provincial strategy, with a massive, resource intensive COVID-19 response that marshalled the entire public health system to prevent infections, contain the pandemic and save lives. These developments are in line with earlier recommendations of the WHO to adopt a whole-of-government approach, namely joined-up activities by different government departments and/or public sector agencies, to ensure coherence in the pandemic response [63].

Contrary to the negative predictions that African countries might not be able to cope with the pandemic [27], our case study found numerous innovations in Gauteng's COVID-19 response. A significant injection of financial resources from the province's own coffers enabled additional staff appointments at hospitals, infrastructure development or refurbishment, and digital innovation and partnerships. The allocation of additional resources has been a common feature of the COVID-19 response in high-income countries [20]. Case studies in LMICs have reported resource allocation from central governments, rather than from state or provincial governments [24, 64]. The availability of dedicated provincial resources for the COVID-19 response shows the relative advantage of Gauteng compared to other South African provinces [44], as well as the positive relationship between densely populated, urban regions, such as Gauteng, and innovation, especially during times of crises [65].

Notwithstanding these innovations, the key informant interviews revealed multiple narratives of contestations and contradictions in the pandemic response. Strong central coordination is needed in times of public health crises and disease outbreaks [63, 66]. However, the interviews revealed the tensions in intergovernmental relations, which predate the COVID-19 pandemic [67]. There were also problems created by a hierarchical, top-down approach, which reportedly did not incorporate or consider the views of health facility managers, frontline staff, or other important stakeholders. These tensions resulted in inappropriate or nonsensical regulations and guidelines, which negatively influenced the acceptance or implementation of the COVID-19 response at health facility level. Another South African study also found that strong local leadership could not counter weak intergovernmental relations, which impacted negatively on the implementation of a major health reform initiative and threatened its sustainability [67]. Notwithstanding different contexts, the negative consequences of poor intergovernmental relations and fragmentation on the COVID-19 response during the first epidemic wave were also found in the federal systems of Italy [12], Spain [16] and the United States [13].

The digital innovation and partnerships that produced provincial modelling of COVID-19 estimates and resource requirements were novel in Gauteng. The key informant interviews revealed the contestations about the modelling process and the results of the modelling, which produced an overestimate of the expected number of COVID-19 cases during the first epidemic wave and hence hospital bed and financial resource requirements. These estimates influenced subsequent provincial decisions (e.g. field hospitals) that aimed to address the gap between bed needs and bed availability. Although the pandemic uncertainty and fear of the unknown also influenced the estimates, the accuracy of modelling is dependent on the robustness and quality of the existing health information system and its utilisation for decision-making. Many key informants highlighted the vulnerabilities of the existing health information systems in the province. These vulnerabilities partly explain the modelling inaccuracies and the lack of agility in the response. Researchers in India have underscored the importance of a seamless integration between a digital disease surveillance system and the existing health information system, the latter an essential component of a resilient health care system [68].

Our case study found that the COVID-19 pandemic exposed and amplified the fragilities and fault lines of the public health care system in Gauteng. A major weakness of the COVID-19 health response in Gauteng was the collateral damage caused by the virtual shut-down of the public health care system for essential health services, the impact of which may only be felt in years to come. Although the disruptions of essential health services appear to be universal [69], other South African scholars have also highlighted the adverse consequences of the suspension or scale-down of essential services [36, 38, 70], and the potential of the reversal of the gains made in mortality reductions and improvements in life expectancy over the past decade [35]. A Gauteng study using routine health information found that both PHC and family planning utilisation declined significantly during the lockdown period [70]. As the ability to deliver essential services is dependent on baseline capacity of the health system [36], the collateral damage to health services is likely to affect poor people disproportionately who are dependent on the public health care system.

The roof of our conceptual framework underscored the criticality of leadership, management and governance. Notwithstanding the innovative central warehousing and a dedicated budget for PPE, the Special Investigating Unit found that the Gauteng Department of Health awarded many PPE contracts irregularly, with unit prices artificially inflated by between 211% and 542% [59]. The reported PPE corruption is a reflection of a legacy of sub-optimal health leadership, management and governance, exacerbated by a prevailing culture of poor accountability, lack of responsibility and unethical behaviours [71, 72].

The key informants pointed to heartwarming examples of health facility leadership, teamwork and frontline health professionals going beyond the call of duty and rising to the challenge of providing health care during the first wave of the pandemic. Further research is needed on what factors account for these positive examples of distributed leadership at local level and staff agency, so that the unintended positive consequences of the pandemic are maximised. However, there was under-investment and insufficient focus on the health workforce, the COVID-19 response failed to take into account or deal with their fears, and to incorporate strategies for psychosocial support, and safe working environments. The under-investment in the health workforce is a global challenge [73], but the COVID-19 pandemic underscored pre-existing capacity and management weaknesses [74, 75] and amplified the chronic under-investment and relative neglect of health workers in the South African health system [37].

Our case study found that there were missed or wasted opportunities to invest in primary health care as the foundation of the health system, partner with communities and civil society, and to explore relationships or collaboration with the private health sector. These missed opportunities reflect longstanding and unresolved areas of contention. For example, the

National Disaster Management Act makes provision for the state to make regulations or issue directions for the purpose of assisting and protecting the public, providing relief to the public and/or dealing with the effects of a disaster [34]. However, government did not consider any regulations to ensure greater equity between the public and private health sectors or one national response.

Our qualitative case study is limited by its cross-sectional nature, as it was conducted at the end of the first epidemic wave. Key informants may have provided different perspectives if we interviewed them during or after the second wave. However, the validation of six interview transcripts and the study results confirm the relevance of our study. We only interviewed three key informants from the private health sector, and this is a study limitation. Although Gauteng province is of strategic importance to South Africa, the results cannot be generalised to the other eight provinces.

Our study has numerous strengths. Firstly, this is one of the first case studies to explore the COVID-19 health system response at a provincial level in South Africa. The adapted Pandemic Emergency Response Conceptual Framework is a useful, analytical tool to explore the COVID-19 response. A methodological strength is the combination of key informant interviews and the document analysis. We obtained rich narratives on the COVID-19 response in Gauteng province, which adds to the discourse on the notion of a resilient health system, able to withstand infectious disease outbreaks and ensure quality universal health coverage. Our case study has highlighted the contestations and contradictions in the responses of key informants and how they perceived or experienced the COVID-19 response in Gauteng. The prevalence of these contradictory perceptions and experiences could be explored in subsequent quantitative studies. Lastly, our study provides lessons for the management of future health crises.

There are several recommendations that arise from this case study. In line with our conceptual framework, the immediate actions should include improved health leadership, management and governance; surveillance, containment and control; avoid collateral damage of the health care system and maintain essential services, prioritise the health workforce; and invest in health information systems.

The COVID-19 pandemic has highlighted the importance of strengthening the disaster management system in the country, with enhanced surveillance and response capacity, able to detect and respond swiftly and effectively. However, an immediate response to a pandemic or crisis cannot be separated from the long-term strengthening of the health care system in Gauteng. Within the critical domain of leadership, management and governance, some issues to discuss or focus on in the medium to long-term, include management structures that are based on function, and consisting of team members selected on merit, and with the authority to manage complex change and build strong systems. Such systems should detect fraud and corruption early on.

Although strong systems are essential, changing the organisational culture is equally important. This will require investment in people, confronting dysfunctional or weak management, fostering staff agency and accountability, and rewarding ethical and professional conduct. Our case study also suggests the importance of the creation of a learning health system to reflect on and learn from mistakes, and that encourages diverse views.

A more decentralised approach, with appropriate delegation of authority, accountability, and consequence management, and reducing the gap between policy and implementation would complement both system change and health workforce strengthening. Open, transparent and seamless communication across government spheres will go a long way in improving intergovernmental relations and ensuring a coordinated, unified response.

In Gauteng, the private health sector is large and prominent. Government has an important stewardship role [76] that includes appropriate legislation or regulation (e.g. costs of tests) and strategies that use a combination of incentives and penalties. Gauteng could lead the way in engaging and partnering with the private health sector in testing different health care delivery models in preparation for the proposed national health insurance system, the country's vehicle for universal health coverage.

The COVID-19 pandemic has challenged the capacity of the state to deliver. In the medium term, there should be an analysis of existing disaster management capabilities in Gauteng and long-term investments needed (competencies, skills, finances). Importantly, the pandemic has demonstrated that a capable state also requires investment in communication systems, including appropriate use of technology, building confidence and trust in government's ability to provide stewardship and leadership of complex changes, to the benefit of the population at large.

A bottom-up approach, investing in and partnering with communities, civil society and other stakeholders will strengthen relationships, ensure ownership of solutions and reduce resistance to change. The COVID-19 crisis provides the opportunity for Gauteng to lead and build on its whole-of-government approach, and to leverage the multi-sectoral and interdepartmental structures and systems for the long-term benefits of improved health system performance and enhanced population health.

## Supporting information

**S1 Appendix. Interview schedules.**
(PDF)

## Acknowledgments

We thank the staff in the Office of the Premier, Mr Mduduzi Mbada and Mr Ismail Akhalwaya for their overall stewardship of the Gauteng case study, and for facilitating the interviews with executive managers in provincial and local government. We acknowledge the support and assistance of Neo Thathe, Kedifentse Merafe and Sanitha Naidoo.

We thank the key informants for the time spent in interviews, and their willingness to share rich insights, information and their experiences with the research team. The findings and conclusions in this article are those of the authors and do not necessarily represent the official position of the Gauteng Provincial Government or its staff.

## Author Contributions

**Conceptualization:** Laetitia C. Rispel, Carol Marshall, Busisiwe Matiwane, Immaculate Sabelile Tenza.

**Data curation:** Laetitia C. Rispel, Carol Marshall, Busisiwe Matiwane, Immaculate Sabelile Tenza.

**Formal analysis:** Laetitia C. Rispel, Carol Marshall, Busisiwe Matiwane, Immaculate Sabelile Tenza.

**Funding acquisition:** Laetitia C. Rispel.

**Methodology:** Laetitia C. Rispel, Carol Marshall, Busisiwe Matiwane, Immaculate Sabelile Tenza.

**Project administration:** Laetitia C. Rispel.

**Resources:** Laetitia C. Rispel.

**Supervision:** Laetitia C. Rispel.

**Validation:** Laetitia C. Rispel.

**Visualization:** Laetitia C. Rispel.

**Writing – original draft:** Laetitia C. Rispel.

**Writing – review & editing:** Laetitia C. Rispel, Carol Marshall, Busisiwe Matiwane, Immaculate Sabelile Tenza.

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
