## [Decision Letter · Decision Letter 0]

15 Sep 2021

PONE-D-21-14402Innovations, contestations and fragilities of the health system response to COVID-19 in the Gauteng Province of South AfricaPLOS ONE

Dear Dr. Rispel,

Thank you for submitting your manuscript to PLOS ONE. After careful consideration, we feel that it has merit but does not fully meet PLOS ONE’s publication criteria as it currently stands. Therefore, we invite you to submit a revised version of the manuscript that addresses the points raised during the review process.

We look forward to receiving your revised manuscript.

Kind regards,

Nora Engel

Academic Editor

PLOS ONE

Journal Requirements:

2. When reporting the results of qualitative research, we suggest consulting the COREQ guidelines or other relevant checklists listed by the Equator Network, such as the SRQR, to ensure complete reporting (http://journals.plos.org/plosone/s/submission-guidelines#loc-qualitative-research). In this case, please consider including more information on the number of interviewers, their training and characteristics. Moreover, please provide the interview guide used as a Supplementary file.

Reviewers' comments:

Reviewer's Responses to Questions

**Comments to the Author**

1. Is the manuscript technically sound, and do the data support the conclusions?

Reviewer #1: Partly

Reviewer #2: Yes

2. Has the statistical analysis been performed appropriately and rigorously? 

Reviewer #1: N/A

Reviewer #2: N/A

3. Have the authors made all data underlying the findings in their manuscript fully available?

Reviewer #1: No

Reviewer #2: No

4. Is the manuscript presented in an intelligible fashion and written in standard English?

Reviewer #1: Yes

Reviewer #2: Yes

5. Review Comments to the Author

Reviewer #1: The interview data was not provided given confidentiality concerns. Sound ethical processes were used to obtain consent etc.

The paper portends to speak about the health sector but in effect speaks to the public health sector only; This is a challenge given that a large private health sector exists in the province and a significant portion of the population uses this sector.

As is usually the case key informants dont always have have same views and opinions (largely depends on where they are positioned in the system). As a consequence there were many contradictory views which leaves the reader uncertain to who reflects the reality! To assist the reader more quantitative data could have been used to support these views.

Reviewer #2: General comments:

• This paper is well written, easy to follow and well-edited – a joy to read!

• The results are really interesting, and the value of the paper, as an in-depth description of a health system response to a shock is valuable.

• The recommendations are apt, and will be of value in the pandemic response moving forward.

• The paper would be strengthened by either a clearly stated research question or a broader argument about what the authors wanted to learn from the exploration of the HS response. I definitely think there is value in a purely descriptive study, but also think that the reader’s attention would be better directed by some indication of what specifically the authors wanted to learn.

• I feel that the sources of data could be better integrated – particularly the document analysis with the interview materials. Firstly, the document analysis is referred to explicitly in the discussion on Theme 3, but not Themes 1, 2, 4 or 5. Were there no contradictions evident in the documents, or innovations, for example?

• Secondly, the epidemiological data/ incidence rates does not seem to have informed the findings as presented in the results section. The authors state that this data was collected to “contextualise interview and provide a visual representation of the pandemic.” However, I would have thought this data was available, and would not need to be collected or analysed by the authors as part of this study. I wonder why that data was included. Perhaps it would simplify the paper to position the epidemiological data as background/contextual information rather than as a core finding of the study, especially since the findings from the interviews and document analysis don’t seem to be triangulated with or interpreted in light of this data.

• Relatedly, a couple of times in the results section, it’s not quite clear whether the finding is supported by the document analysis, interview data or epidemiological data (or all three). For example, ln 662-3, and 711-12

Abstract

• Pg 2, ln 38-39: The first line of the results section of the abstract feels more like a value judgement than a presentation of findings, and also feels at odds with the rest of the findings as presented in the abstract. I realise that this is one of the themes of the findings, but perhaps consider making that clear in the phrasing/structure of the paragraph: i.e. perhaps rephrase to ‘innovation and multi-sectoral collaboration was evident’ or similar?

Introduction

• Pg 4, ln 75 – Consider reporting excess deaths instead/in addition, to paint a clearer picture?

• Pg 4 ln 70-76: Consider moving this paragraph down, so that the structure of the introduction goes from a broad, global-level discussion, and then introduces Africa, and South Africa more specifically. I.e. move to page 7, ln 127. It might also help to start the discussion of the South African case with a background paragraph that gives the reader the information they need about the political (e.g. quasi-feralism), economic and social context in South Africa. Could also include where the responsibility for health decisions lies. This information could also be included in ‘study setting.’

• Pg 5, ln 90: I would include a ‘at time of writing’ caveat before this paragraph. Because, for example, Australia is currently seeing dramatic increases.

• Pg 7, ln 131: I suggest the reader needs more information here – i.e. ‘resulted in corruption scandals including, for example….’ The reference to ‘the reported corruption’ suggests a) that there is only one, and that the reader already knows what it is.

• Pg 7, starting ln 136: This paragraph is doing 2 different things at one: firstly, it justifies the choice of Gauteng province as the case study, and then it justifies the study as whole (with reference to the shortage of empirical studies on this topic). I suggest, as above, that while the former justification is clear, the latter needs a paragraph of its own, in which the authors specify the phenomenon of interest more precisely.

• Ln 132-134: Feels out of place in this paragraph – as above I suggest introducing a paragraph on background of SA for this type of information

• Ln 143: Include Guateng proportion of population, so that the reader can interpret the weight of the percentage of cases.

• Pg 9, ln 184: This is a bit confusing because saving lives and prevention of deaths reads like an overall goal of the pandemic response, rather than one of the pillars. You description of it, starting at ln 193 makes sense, so, I wonder whether the issue is just the ‘naming’ of the pillar. Would something like ‘service provision and treatment’ make more sense?

• Pg 11, ln 229 – It’s not quite clear why the ‘framing’ of policies was important to understand, and furthermore, the issue of framing isn’t raised again in the paper.

• Pg 12, ln 229 – Because the phenomenon of interest for this study is not well defined, the reader wonders why it was important to understand the ‘framing of the policy issues’

• Pg. 12, ln 230: Consider rephrasing to “to allow for triangulation of various sources of data”. It doesn’t make sense to say that you did document analysis to triangulate data from the document analysis.

• Pg 16, data analysis: It’s not quite clear how the researchers assessed completeness prior to the analysis. In other words, without starting the analysis, how did the researchers know that data saturation had been reached? Is this perhaps more a point about data collection?

• Pg 17, ln 325 – just for clarity was the consensus reached about the themes being appropriate and sufficient, i.e. that no new themes were emerging? Or that the researchers all agreed on the themes?

• Pg 18, ln 351: “and the documents analysed” – does this refer to triangulation between the interview data and the document review? The phrasing is a bit unclear.

Results

• Pg 39, ln 651: You write that “the central warehousing and procurement of PPE were a Gauteng innovation” – should this not then be referenced as a finding from the document analysis under theme 1?

Discussion

• Pg 48, ln 853 – “At face value” – I wonder whether the positioning here is that the Gauteng response was ‘at face value’ innovative, comprehensive and impressive, or whether it is more appropriate to the data to say that the response was comprehensive and in some ways impressive, but that while some innovations were evident, there were also significant challenges and shortcomings. I feel that you don’t want to imply that the innovations were illusory.

6. PLOS authors have the option to publish the peer review history of their article (what does this mean?). If published, this will include your full peer review and any attached files.

Reviewer #1: **Yes: **Yogan Pillay

Reviewer #2: No

---

## [Author Response · Author response to Decision Letter 0]

12 Nov 2021

RE: RESPONSE TO REVIEWERS-PONE-D-21-14402

Title: Innovations, contestations and fragilities of the health system response to COVID-19 in the Gauteng Province of South Africa

We thank the editor and the reviewers for the comments on the above-mentioned manuscript that we received five months after submission, with the delay of a further 2 weeks on the caption for the supporting information. 

In the cover letter, we have included a table with the comments from the editor and the reviewers, our detailed responses as authors, as well as the page and line references where the changes were made. The document is 13 pages long. We have included a manuscript with changes highlighted, as well as a clean version of the manuscript.

---

## [Editor Report · Decision Letter 1]

1 Dec 2021

Innovations, contestations and fragilities of the health system response to COVID-19 in the Gauteng Province of South Africa

PONE-D-21-14402R1

Dear Dr. Rispel,

We’re pleased to inform you that your manuscript has been judged scientifically suitable for publication and will be formally accepted for publication once it meets all outstanding technical requirements.

Kind regards,

Nora Engel

Academic Editor

PLOS ONE
---

## [Editor Report · Acceptance letter]

9 Dec 2021

PONE-D-21-14402R1 

Innovations, contestations and fragilities of the health system response to COVID-19 in the Gauteng Province of South Africa 

Dear Dr. Rispel:

I'm pleased to inform you that your manuscript has been deemed suitable for publication in PLOS ONE. Congratulations! Your manuscript is now with our production department. 

Kind regards, 

on behalf of

Dr. Nora Engel 

Academic Editor

PLOS ONE